# Chitosan-Urea Nanocomposite for Improved Fertilizer Applications: The Effect on the Soil Enzymatic Activities and Microflora Dynamics in N Cycle of Potatoes (*Solanum tuberosum* L.)

**DOI:** 10.3390/polym13172887

**Published:** 2021-08-27

**Authors:** Rohini Kondal, Anu Kalia, Ondrej Krejcar, Kamil Kuca, Sat Pal Sharma, Karanvir Luthra, Gurmeet Singh Dheri, Yogesh Vikal, Monica Sachdeva Taggar, Kamel A. Abd-Elsalam, Carmen L. Gomes

**Affiliations:** 1Department of Microbiology, Punjab Agricultural University, Ludhiana 141004, Punjab, India; rohini.kondal13@gmail.com (R.K.); luthrakaran11@gmail.com (K.L.); 2Electron Microscopy and Nanoscience Laboratory, Department of Soil Science, Punjab Agricultural University, Ludhiana 141004, Punjab, India; 3Center for Basic and Applied Science, Faculty of Informatics and Management, University of Hradec Kralove, 50003 Hradec Kralove, Czech Republic; ondrej.krejcar@uhk.cz; 4Malaysia Japan International Institute of Technology (MJIIT), Universiti Teknologi Malaysia, Jalan Sultan Yahya Petra, Kuala Lumpur 54100, Malaysia; 5Department of Chemistry, Faculty of Science, University of Hradec Kralove, 50003 Hradec Kralove, Czech Republic; 6Department of Vegetable Science, Punjab Agricultural University, Ludhiana 141004, Punjab, India; sharmasp@pau.edu; 7Green House Gas Laboratory, Department of Soil Science, Punjab Agricultural University, Ludhiana 141004, Punjab, India; gsdheri@pau.edu; 8School of Agricultural Biotechnology, Punjab Agricultural University, Ludhiana 141004, Punjab, India; yvikal-soab@pau.edu; 9Department of Renewable Energy Engineering, Punjab Agricultural University, Ludhiana 141004, Punjab, India; monicasachdeva@pau.edu; 10Agricultural Research Center, Plant Pathology Research Institute, Giza 12619, Egypt; kamelabdelsalam@gmail.com; 11Department of Mechanical Engineering, Iowa State University, Ames, IA 50011, USA; carmen@iastate.edu

**Keywords:** chitosan, nanocomposite, nanofertilizer, slow release, ammonia oxidase gene, quantitative polymerase chain reaction, microflora N cycle, nutrient use efficiency, soil N content

## Abstract

The impact of polymer-based slow-release urea formulations on soil microbial N dynamics in potatoes has been sparingly deciphered. The present study investigated the effect of a biodegradable nano-polymer urea formulation on soil enzymatic activities and microflora involved in the N cycling of potato (*Solanum tuberosum* L.). The nano-chitosan-urea composite (NCUC) treatment significantly increased the soil dehydrogenase activity, organic carbon content and available potassium compared to the conventional urea (CU) treatment. The soil ammonical nitrogen (NH_4_^+^-N) and nitrate nitrogen (NO_3_^−^-N) contents and urease activity were significantly decreased in the NCUC-amended soil. The slow urea hydrolysis rate led to low concentrations of NH_4_^+^-N and NO_3_^−^-N in the tested potato soil. Furthermore, these results corroborate the low count of ammonia oxidizer and nitrate reducer populations. Quantitative PCR (q-PCR) studies revealed that the relative abundance of eubacterial (AOB) and archaeal ammonia-oxidizing (AOA) populations was reduced in the NCUC-treated soil compared to CU. The abundance of AOA was particularly lower than AOB, probably due to the more neutral and alkaline conditions of the tested soil. Our results suggest that the biodegradable polymer urea composite had a significant effect on the microbiota associated with soil N dynamics. Therefore, the developed NCUC could be used as a slow N-release fertilizer for enhanced growth and crop yields of potato.

## 1. Introduction

Potato (*Solanum tuberosum* L.) is one of the most abundant and widely cultivated vegetable crops in the world. It is the third most important crop after rice and wheat, with an annual production of 376 million tons (FAOSTAT, 2018). The growth and yield of potato strongly depend on the availability of nutrients, especially nitrogen (N) [1]. Urea is widely used as a N source because it can trigger biomass and total N accumulation in potatoes [2]. Potato requires high doses of N fertilizers to achieve maximum yield; however, N recovery in potatoes is often low because of the plant’s poorly developed root system [3]. Therefore, the input of N fertilizers has been increased over the years to maximize tuber yields; however, excessive application of N fertilizers may reduce tuber yields [4]. In addition, it also significantly influences N cycling and augments NH_3_^+^ emissions (volatilization) and NO_3_^−^-N accumulation (leaching), which have serious environmental implications [5]. This calls for the development of effective N management strategies such as controlled-release fertilizers (CRF) to improve N use efficiency and also to combat the environmental impact of fertilizer application [6].

Polymer-coated urea (PCU) is a urea fertilizer formulation that exhibits a more predictable release pattern. It enables the nutrient release to synchronize with the needs of the plant [7,8]. Recently, the use of controlled-release urea (CRU) has become a new trend to enhance crop yields due to its high potential for enhancing nutrient use efficiency (NUE) [9]. A study has shown that PCU in cotton can significantly affect the NH_4_^+^ and NO_3_^−^ concentrations in the soil by lowering them as compared to conventional nitrogen fertilizers [10]. Another study on PCU reported that NH_3_ volatilization was significantly reduced when compared to surface-applied urea on creeping bent grass [11].

The concentrations of NO_3_^−^ and NH_4_^+^ in soils are controlled by various factors, such as soil temperature, pH, soil microbiota, fertilizer form and moisture conditions [12]. The biological component of soils usually responds to the changing soil conditions more rapidly than either the chemical or physical properties [13]. Soil enzymatic activities have been used as indicators of microbial activity because these activities reflect the total range of oxidative activity in soil microflora [14,15]. Previous studies on dehydrogenase activities in maize revealed that a polyolefin-coated CR urea fertilizer formulation significantly increased the soil microbial activity compared to conventional urea [16]. However, the urease enzyme activity was decreased when different biodegradable urea coating formulations were incorporated in comparison to conventional urea, leading to slower urea hydrolysis rates [17].

Previous studies focused on improving the slow release properties of chitosan have been carried out [18,19]. However, these studies did not focus on nitrogen release properties. Likewise, few other published reports involved evaluation of the effect of controlled-release urea fertilizers in potato, but these studies were on the use of polymer sulfur or polyurethane as a coating material and not chitosan [20]. The benefits of the use of chitosan as a urea-encapsulating/embedding agent are two-fold. The primary benefit is that the chitosan polymer has a biological origin [21] and is bio-safe and biodegradable compared to synthetic polymers which cannot be metabolized by the soil microbes [22]. Further, it has already been utilized for immobilization of enzymes for food [23], and biosensing [24,25] applications. The second benefit compared to inorganic coatings such as sulfur coatings is the absence of cracks and the lowered ability to cause acidification of the soil [26,27]. Furthermore, there are no reports published that discern the effect of different chitosan concentrations on the controlled release properties of nitrogen nanofertilizers for use as N fertilizers in potato. In this study, the hypothesis tested stated that the nano-chitosan-urea composite (NCUC) will have a significant effect on the microflora involved in N cycling. This formulation may also decrease the ammonical N and nitrate N content in the soil. A pot experiment was conducted using a nano-chitosan-urea composite in comparison to conventional urea at four different N application rates (0, 50, 75 and 100% recommended N dose per pot). The primary objectives of this study were to determine the effect of NCUC on (1) soil microbial viable counts and soil enzymatic activities, (2) soil chemical properties and (3) soil ammonia-oxidizing eubacterial and archaea-bacterial populations involved in the N cycling of potato cv. Kufri Pukhraj.

## 2. Materials and Methods

### 2.1. Materials

Chitosan (medium molecular weight: 190 to 310 kilo Daltons, and low DDA percentage (>75%), 100% purity) was purchased from Sigma-Aldrich, St. Louis, USA. Sodium tripolyphosphate (purity: 98.0%) was also procured from Sigma Aldrich, St. Louis, USA. Analytical-grade urea (purity: 99.5 to 100%) was purchased from HiMedia Laboratories Pvt. Ltd., Mumbai, India. HPLC-grade water was utilized for the preparation of all the formulations. The other analytical-grade chemicals utilized for the determination of the various enzyme activities were purchased from Hi-Media Laboratories Pvt. Ltd., Mumbai, India, while the consumables required for the q-PCR studies were purchased from DSS Takara BioIndia Pvt. Ltd., New Delhi, India.

### 2.2. Synthesis and Characterization of Nano-Polymer Urea Composite

Chitosan (1.5% *w*/*v*) was dissolved in acetic acid (1.0% *v*/*v*) containing Tween 80 as a surfactant to prevent the agglomeration of the nanoparticles during and after synthesis. Urea (analytical grade, 10% *w*/*v*) was dissolved in the chitosan-acetic acid suspension. Under constant stirring conditions, the aqueous tripolyphosphate (TPP) solution was added drop wise (16 drops per minute) to chitosan-urea suspensions under room temperature conditions for gelation. Instantaneously, the synthesis of chitosan-urea nanoparticles started due to the TPP-initiated ionic gelation mechanism [28,29,30,31]. The coalescence of the formed nanoparticles was reduced by ultrasonication of the formulation using a probe sonicator (VCX-750, Sonics and Materials Inc., Newtown, CT USA) operated at 35% amplitude for 15 min under ice bath conditions to avoid over-heating of the formulation. The chitosan-urea formulations were subjected to freeze drying in a lyophilizer assembly (REVA commercial lyophilizer, Gujarat, India) at a pressure of 10^−1^ torr (at −40 °C) for 24 h to obtain nano-chitosan-urea composite (NCUC) powdered samples.

### 2.3. UV-Vis Spectroscopy

The chitosan-urea composite formulation was analyzed for its unique light matter interaction using a double beam UV-Vis spectrophotometer (model SL 218, ELICO, Hyderabad, India) for wavelengths ranging from 190 to 800 nm. The absorbance was plotted against the wavelength to observe the characteristic UV-Vis absorbance peaks for free urea, the chitosan-TPP composite (control) and the chitosan-urea nanocomposite.

### 2.4. Transmission Electron Microscopy (TEM)

The specimens for TEM analysis were prepared by dispensing a known amount of freeze-dried powdered NCUC in deionized distilled water. These suspensions were sonicated for 15 min in a bath sonicator (model SW-4, Toshcon, India). A known aliquot of the sonicated sample suspension was placed on a 200 mesh size carbon-coated copper grid (Tedpella, Redding, CA, USA). The sample was allowed to adsorb and form a thin film on the carbon coating by incubating for 2 to 3 min. The grids were then allowed to air dry at room temperature overnight. The dried grids were imaged in high-resolution imaging mode using a transmission electron microscope (model Hitachi H-7650, Hitachi High-Technologies Corporation, Tokyo, Japan) at an 80 kV accelerating voltage. The size dimensions of the chitosan and chitosan-urea nanoparticles were measured through Image J software (version 1.46 r, U. S. National Institute of Health, Bethesda, MD, USA) by manually obtaining the diameters for >50 particles from five TEM micrographs.

### 2.5. Scanning Electron Microscopy (SEM) and SEM-Energy-Dispersive Spectroscopy (SEM-EDS)

The morphology and cross-sectional topography of the synthesized NCUC were studied by a scanning electron microscope (model Hitachi s-3400N, Hitachi High-Technologies Corporation, Tokyo, Japan) operated at a 15 kV accelerating voltage in secondary electron imaging mode. The freeze-dried NCUC was placed on the stub and sputter coated for 30 s (current: 18–20 mA) with gold plasma in an ion sputter coater (model Hitachi E-1010, Hitachi High-Technologies Corporation, Tokyo, Japan) to make the sample conductive. The elemental composition, the percentage of atoms and the weight of elements present on the sample surface were analyzed by SEM-EDS (model Thermo Noran, Thermo Fisher Scientific Inc., Waltham, Massachusetts, USA).

### 2.6. Fourier Transform-Infrared Spectroscopy (FT-IR Spectroscopy)

FT-IR spectroscopy for the freeze-dried powdered samples of the NCUC was performed. The samples were mixed uniformly with pre-activated potassium bromide in an appropriate ratio (1:100), and pellets were prepared by using a hydraulic pellet press assembly (Model no. 1701, Maharashtra, India). The prepared pellets were placed in the sample mounting region of the FT-IR spectroscope (model Thermo Nicolet 6700, Thermo Fisher Scientific Inc., Waltham, Massachusetts, USA) and were scanned in the mid-IR range of 400–4000 cm^−1^ to obtain transmission spectra at a 4.0 cm^−1^ spectral resolution using 32 scans.

### 2.7. Urea Release Profile and Encapsulation Efficiency

The release of the encapsulated urea from the nano-chitosan-urea composite was performed by extracting aliquots (1 mL) at different time intervals (after every five days) from 500 mL autoclaved deionized water containing known concentrations (1, 10 and 100 mg per liter) of nano-chitosan-urea composites (sink conditions). The aliquots drawn were evaluated for urea content (in mg L^−1^) using the acidic p-dimethyl aminobenzaldehyde method [32] by measuring the absorbance of the solution at 422 nm against a reagent blank in a dual beam UV-Vis spectrophotometer (model SL-218, ELICO, Hyderabad, India). At each time interval, the sample aliquot was filtered using a 0.2 µm nylon membrane syringe filter and then analyzed spectrophotometrically at 422 nm. For the encapsulation efficiency, a known amount of NCUC was crushed and dispersed in distilled water, and then left in suspension for 72 h at 400 rpm at room temperature. The suspensions were passed through a 0.2 µm polypropylene syringe filter (VWR Intl., Radnor, PA, USA) to remove the chitosan. The amount of urea was determined by using the acidic p-dimethyl amino benzaldehyde method [32]. The entrapment efficiency was calculated according to Equation (1).
(1)$$EE\left(\%\right)=\frac{amount\;of\;active\;compound\;entrapped}{initial\;active\;compound\;amount}*100$$

### 2.8. In Vitro Biodegradation Assay

The biodegradability of the NCUC was evaluated through an in vitro biodegradation assay by incubating a known pre-weighed quantity of the sample in phosphate-buffered saline (2 mL, pH = 7.4) and a lysozyme mixture (Hen egg white origin, Sigma Aldrich, St. Louis, MO, USA, 1 mg mL^−1^) followed by weighing of the incubated sample retrieved after 1, 3 and 7 days of incubation. Both the initial and retrieved samples were freeze dried to get rid of the water molecules. The biodegradation was calculated as
Biodegradation (%) = W_1_ − W_2_/W_1_ ∗ 100
where W_1_ = initial weight of NCUC, and W_2_ = weight of retrieved NCUC.

### 2.9. Experimental Design

The soil (topsoil 0–20 cm) used in this study was collected from fields of the Vegetable Farm in the Department of Vegetable Science, Punjab Agricultural University, Ludhiana, Punjab, India. The collected sandy loam soil (Dystric Cambisol) was sieved through a 2 mm iron mesh. Then, the plastic pots (30 cm height and 11 cm diameter, 8 kg capacity) were filled with 5 kg of soil. The experiment was carried out from October 2016 to March 2017 in open conditions with average temperatures ranging between 23 and 34.6 °C and relative humidity of 75%. The experimental tuber seeds of *Solanum tuberosum* L. (potato) cv. Kufri Pukhraj were also obtained from the Department of Vegetable Science. Each pot was watered 24 h before sowing the tuber, and sowing was conducted with one tuber per pot. The recommended dose of nitrogen was applied at the rate of 75 kg N or 165 kg urea acre^−1^. All three types of fertilizer, the nano-chitosan-urea composite (NCUC), conventional urea (CU) and chitosan (CS), were applied at four different concentrations of 0, 50, 75 and 100% of the recommended nitrogen fertilizer dosage (75 kg acre^−1^). The phosphorus and potassium fertilizers were applied at a rate of 25 kg acre^−1^. The soil samples were collected at 0, 20, 40, 60 and 90 days after treatment (DAT) and subjected to various analyses (described below).

### 2.10. Soil Chemical Characterization

The soil samples taken at different time intervals were dried in an oven at 60 °C for 2 days or until a constant weight of the samples was achieved. The dried soil was grinded in a pestle and mortar and sieved through a 2.0 mm sieve to get rid of any organic or root debris. The dried and sieved soil samples were used for chemical property characterization. The pH and electrical conductivity (EC, soil/water ratio of 1:2) of the soil samples were determined by the glass electrode conductivity method on pH (model μpH system 361, Systronics India (Pvt.) Ltd., Ahmedabad, Gujarat, India) and a solubridge meter [33]. Soil organic carbon (OC) was estimated by following the procedure described by Walkley and Black (1934), which involved the use of a soil (2.0 g) sample which is reacted with 1 N potassium dichromate (100 mL)+20 mL sulfuric acid+NaF (0.5 g)+Diphenylamine indicator solution followed by titration of 10 mL of this mixture against 0.5 N ferrous ammonium sulfate solution. The macronutrient status (N, P and K) of the soil was determined using methods described by Kjeldahl [34], and Reed and Scott [35]. The soil P content was measured through the ammonium molybdate technique, while the K content was estimated through flame photometry. The available ammonical nitrogen and nitrate nitrogen in the soil were determined by the Kjeldahl technique and included mixing the soil sample (10.0 g) in KCl (2N, 100 mL) solution. The contents were shaken (1 h), filtered through filter paper (Whatman no. 1 filter paper, Kent, U.K.) and analyzed. For ammonical N estimation, sulfuric acid (N/200, 5.0 mL) and methyl red indicator (50.0 μL) were mixed, and the delivery tube end of the Kjeldahl flask was dipped in it. The soil extract (10.0 mL) was incubated with MgO powder (0.2 g) in a distillation flask, and the collected distillate (30.0 mL) was titrated against NaOH (N/200). The amount of ammonical N was calculated from the volume of sulfuric acid (N/200) used for the absorption of NH_3_. For the nitrate N content estimation, the soil extract (10.0 mL) taken in the distillation flask was mixed with 0.2 g each of MgO and Deverda’s alloy. The distillation process was performed in a similar manner to that described for ammonical N, and nitrate N was calculated.

### 2.11. Enumeration of the Culturable Microbial Population

The soil microbial count was enumerated using the spread plate method [36]. Briefly, ten grams of fresh soil samples was transferred to an Erlenmeyer flask (150 mL) containing 90 mL sterile distilled water and shaken at 120 rpm for 15 min to obtain a homogenous suspension. Serial dilutions (up to 10^−6^) were produced by pipetting 1 mL of the soil suspension into 9 mL of sterile water. Finally, a 0.1 mL aliquot of the diluted soil suspension was uniformly spread with the help of a sterile spreader on solidified Petri plates with nutrient agar for total aerobic bacteria, NH_4_ oxidizer agar for ammonia oxidizers, nitrate agar for nitrate-reducing bacteria and Potato Dextrose Agar (PDA)/Sabouroaud’s Dextrose Agar (SDA) (HiMedia Laboratories Pvt. Ltd., Mumbai, India) for total fungi counts, at 0, 20, 40, 60 and 90 days after sowing (DAS). Dilutions of 10^−5^ and 10^−6^ were selected for the enumeration of bacteria, and 10^−3^ to 10^−4^ for fungi, ammonia oxidizers and nitrate-reducing bacteria based on their population in the soil. The Petri plates were incubated for 2 to 6 days at 28 ± 2 °C. The total aerobic bacterial count of the soil samples was enumerated by plating the 10^−5^ and 10^−6^ dilutions on nutrient agar, and the number of bacterial colonies appearing after 24 to 48 h of incubation was counted. The nitrate-reducing bacteria were enumerated by counting the microbial colonies that grew on nitrate-reducing media. Two plates per dilution of each soil sample were plated, counted and reported in CFU g^−1^ of soil.

### 2.12. Soil Enzyme Activity

The effect of different N fertilizer treatments was observed on two specific soil enzyme activities, dehydrogenase and urease. The collected soil was dried and sieved through a 2 mm mesh. The procedures for the dehydrogenase and urease activities were performed following the procedures described by [37,38]. The brief procedure for the dehydrogenase activity that was followed included mixing of CaCO_3_ (0.1 g) in pre-sieved sample soil (10.0 g). The contents were pulverized to obtain a homogenous mixture. A known amount (3.0 g) of the above mixture was mixed with 2,3,5-Triphenyl tetrazolium chloride solution (3% *w*/*v*, 0.5 mL) and distilled water (1.5 mL). The contents were incubated for 24 h at 37 °C followed by methanol (5.0 mL) extraction under shaking conditions to obtain a pink-colored solution after repeated washing of the soil and filtration. The volume was made up to 25.0 mL with methanol, and the absorbance was measured at 485 nm wavelength on the spectrophotometer using methanol as a blank. The protocol used for the estimation of the urease activity of the soil samples involved addition of urea (2000 mg L^−1^, 5.0 mL) to grinded and sieved (sieve size: <2 mm) soil samples (5.0 g). After 5 h of incubation at 37 °C, potassium chloride-phenylmercuric acetate solution (2.0 M, 50.0 mL) was added, and the contents were further incubated for 60 min under shaking conditions. After filtering the contents, the extract (1.0 mL) was mixed with extracting (5.0 mL) and coloring reagents (15.0 mL, Diacetyl monoxime (2.5% *w*/*v*, 25.0 mL)+Thiosemicarbazone (0.25% *w*/*v*, 5.0 mL)). The contents were heated in a boiling water bath for 30 min and cooled downed to room temperature, and the volume was made up to 25 mL using distilled water. The contents were then mixed thoroughly, and the intensity of the red color was measured (λ = 527 nm).

### 2.13. Molecular Characterization of Eubacterial and Archaeal Ammonia-Oxidizing Populations by q-Polymerase Chain Reaction (q-PCR)

Soil samples at 0, 20, 40 and 60 DAT were collected, and isolation of total bacterial community DNA was carried out by HiPurA^TM^ soil DNA purification kit (MB542) purchased from Hi-Media Laboratories (Mumbai, India). DNA quantification was performed on 0.8% gel. Ammonia oxidase gene primers used in this study included the eubacterial ammonia-oxidizing bacteria (AOB) (forward primer: amoA-1F (GGGGTTTCTACTGGTGGT), reverse primer: amoA-2R (CCCCTCTGGAAAGCCTTCTTC)) [39] and archaebacterial ammonia-oxidizing archaea (AOA) (forward primer: amoA310f (TGGATACCGTCAGCAATG), reverse primer: amoA529r (GCAACAGGACTATTGTAGAA)) primers for the ammonia oxidase gene [40], while the 16s rRNA gene universal primer (forward primer: F984 (AACGCGAAGAACCTTAC), reverse primer: R1401 (GGGTCTTGTACACACCG)) was also used. These primers were used for total soil DNA and in all in vitro amplification reactions performed in master cyclers (Eppendorf, Hamburg, Germany and Roche Applied Sciences, Penzberg, Upper Bavaria, Germany) using 14–50 ng genomic DNA of each sample in a final volume of 10 μL of PCR mix.

### 2.14. Potato Vegetative Growth and Yield Traits

The potato plants were uprooted from pots, and the shoot fresh weight per plant was recorded; afterwards, they were oven dried at 60 °C for 2 days, and their dry weight was recorded using an electronic weighing balance in grams at 90 DAS. The yield-attributing characteristics such as the number of tubers formed per plant, total tuber weight per plant and marketable tuber yield per plant were also determined. The marketable tubers were considered by the number and weight of the potatoes above 35 to 40 g [41].

### 2.15. Statistical Analysis

The experiment was carried out in a completely randomized block design with a factorial arrangement. The main plots were fertilizer types (nano-chito-urea (NCUC) and conventional urea (CU)), with the level of fertilizer applications as the sub-plot factors. For all the analyses of this study, determinations were performed in triplicate as independent experiments. Data on different variables were subjected to three-way analysis of variance (ANOVA) with SAS software (Version 9.3, Cary, NC, USA) [42,43], and significant differences between treatments were determined by pair-wise comparisons using Fischer’s least significant difference test (LSD; *p* ≤ 0.05).

## 3. Results

### 3.1. Characterization of Nano-Polymer Urea Composite

The UV-Vis absorbance spectra of chitosan and the nano-chitosan-urea composite are shown in Figure 1D. The chitosan-TPP composite exhibited an absorption peak at 220 nm [44]. An earlier study by Abd-Elhady et al. [44] also reported the occurrence of a specific UV absorption peak for the chitosan nanoparticles to exist at 226 nm. The two UV chromogenic functional groups in chitosan, i.e., N-acetyl glucosamine and glucosamine, exhibit absorption of the far UV wavelength(s). Both of these UV chromogenic groups have an additive role for the absorption peak at 201 nm [45]. However, after incorporation of urea (10% *w*/*v*) into chitosan (1.5% *w*/*v*), dichotomization of the absorption peak at 220 nm occurred with the appearance of a blue-shifted shoulder peak at 200 nm. This indicates a shared occurrence of both the chitosan matrix and chitosan-TPP nanoparticles in the NCUC sample. 

These results were further verified through TEM analysis with size dimensions of the CS-urea NPs embedded in NCUC ranging from 5.67 to 55.67 nm (Figure 1(Ab)). While the overall range of the particle size varied from 69.61 to 154.186 nm in the chitosan sample, the average size of the nanoparticles was 33.39 ± 11.84 and 113.55 ± 19.02 nm in the NCUC and chitosan samples, respectively. The SEM analysis of the 1.5% NCUC (10% *w*/*v*) exhibited an increase in the networked structure of the NCUC, probably due to the formation of rodlets and rodlet bundles within the matrix of the lyophilized product (Figure 1(Bd)). The elemental composition of the prepared NCUC, as depicted by the EDS spectra, showed the presence of three major elements in percentage weight and percentage atom composition, i.e., C, N and O elements (Figure 1(Cf)). The urea incorporation enhanced the nitrogen percentage, as indicated by the corresponding increase in percentage weight and percentage atom content for the nitrogen element (Figure 1(Cf)).

The FT-IR spectra of chitosan showed the presence of characteristic absorption peaks for NCUC (Figure 1E). The spectra demonstrated the occurrence of specific bands at 1550–1650 cm^−1^, 1600–1700 cm^−1^ and 2900 cm^−1^, which corresponded to the individual amide group vibrations [46,47]. The FT-IR analysis (Figure 1E) revealed the occurrence of peaks at 1634 cm^−1^ in chitosan associated with the stretching vibrations of NH-CO and C=O along with bending bonds of N-H and NH_2_ [48], whereas incorporation of urea in chitosan showed sharp and narrow peaks at 1620 cm^−1^ and 1461 cm^−1^, which are due to amide II bending and C–N stretching vibrations [46]. The urea release profiles of NCUC showed a two-step biphasic process (Figure 1F). These results indicate an initial burst effect up to 10–18 days followed by successive slower release up to the 30th day. The 1.5% NCUC, at 10% (*w*/*v*) urea nanoparticle loading, showed an increased release rate of urea compared to the 3 and 5% urea-chitosan formulations. The 1.5% NCUC with 10% urea encapsulated exhibited an 85% encapsulation efficiency. The in vitro biodegradation study involved loss of weight to half by the seventh day for the CS sample (Figure 2). However, the enhanced cross-linking by the addition of urea in the NCUC sample led to a lowered degradation rate compared to the CS sample.

### 3.2. Soil Chemical and Microbiological Properties

The effect of the slow-release nano-chitosan-urea composite fertilizer on different parameters of soil chemical properties is presented in Table 1 and Table 2. The CU recorded the highest (*p* ≤ 0.05) soil pH and EC values followed by CS and NCUC. Among the days after treatment, CU showed a significantly higher soil pH at 60 DAT, whereas the highest EC was observed at 90 DAT (Table 2). On the other hand, the OC showed a different trend, with the CS treatment exhibiting the highest OC value (0.35%) at 90 DAT followed by NCUC and CU (Appendix A).

The soil microbial viable cell populations of total aerobic bacteria, ammonia oxidizers, nitrate-reducing bacteria and fungi in the soil amended with fertilizer showed significant changes among the N fertilizer sources. The count of total aerobic bacteria in the NCUC treatment was significantly higher than that for the CU and CS treatments. On average, the populations of aerobic bacteria in the soil amended with NCUC were 0.11-fold higher than those in the CU treatment. The highest counts of ammonia-oxidizing and nitrate-reducing bacterial populations were observed for CU at 100% concentration-treated soils followed by treatment with NCUC at the same N application level. Contrarily, the CS treatment showed a meager increase in ammonia oxidizers and the nitrate-reducing bacterial count at the 100% N concentration level. The fungal population was the highest in NCUC-treated soils followed by the CU and CS treatments.

The macronutrient status of the soil (i.e., NH_4_^+^-N, NO_3_^−^-N and K) varied among the different fertilizer sources during potato cultivation (Appendix A). The CU recorded significantly higher ammonical N (Figure 3a) followed by NCUC and CS over a period of 60 DAS. After 60 DAS, there was a sudden decline in the concentration among all the fertilizers. A similar trend was observed for the nitrate N contents (Figure 3b). Among the DAS, significantly higher nitrate N contents were observed at 60 DAS followed by a gradual decrease for both CU and NCUC treatments, with a greater decrease for the NCUC treatment. Conversely, the highest K (Figure 3c) content in the soil was recorded for the NCUC treatment at 60 DAT followed by CS and CU treatments.

### 3.3. Soil Ammonia-Oxidizing Microbial Population Analyzed through Real-Time PCR (q-PCR)

The relative abundance of the ammonia-oxidizing eubacterial and archaeal gene (amoA) and the 16S rRNA reference gene in response to two fertilizer sources, CU and NCUC, was analyzed by real-time PCR (q-PCR). The results show that ammonia-oxidizing bacteria exhibited a significantly higher abundance in the CU than in the NCUC treatment, and in particular, there was a 2-fold decrease in the amoA gene abundance in NCUC with 75% and 100% of the recommended dose of the fertilizer treatment at 40 and 20 DAT compared to the respective conventional fertilizer treatment (Figure 4). Among the various treatments applied, the 100% N application level showed the highest relative abundance for both archaeal (AOA) and eubacterial (AOB) ammonia-oxidizing bacterial populations. In comparison, the abundance of AOA was found to be relatively lower than that of AOB. Thus, it seems that the AOB population can be considered the predominant N-transforming microbes affecting the nitrification process in the aerobic sandy loam soils.

### 3.4. Soil Enzymatic Activities

The various sources of fertilizers were tested for soil enzymatic activities. The days after sowing (DAS), the sources (CU, NCUC and CS) and their concentrations exhibited significant differences in the soil dehydrogenase activity. Among the various sources applied, CSNF showed a sudden (*p* ≤ 0.05) increase in dehydrogenase activity followed by NCUC and CU treatments up to 20 DAS, with a slight increase up to 60 DAS (Figure 5a). The highest urease activity was observed in the CU treatment at 60 DAS at the 100% N application level compared to the other treatments (Figure 5b).

### 3.5. Potato Vegetative Growth and Yield Traits

The shoot growth was significantly enhanced by the NCUC application (Figure 6) compared to the conventional urea treatment at an equivalent dose of application. 

The shoot fresh and dry weights of the sampled plants were observed to be highest in the NCUC treatment (Table 3). In addition, a dose-dependent effect on the shoot fresh and dry weight parameters was recorded for both the CU and NCUC treatments.

The tuber yield traits including the tuber number and average tuber weight per plant were also recorded to have improved by application of the nano-chitosan-urea composite. The type of N fertilizer and dose of application exhibited a dual interactive effect (Appendix A) for the above yield traits.

## 4. Discussion

The TEM studies (Figure 1A) revealed that the addition of urea (10% *w*/*v*) in NCUC resulted in a reduction in the nanoparticle sizes, with the size range varying from 11 to 35 nm. This may be due to the pore size of the NCUC matrix, which could have decreased due to the enhanced networking within the matrix. These results were found to be consistent with another study [49] which reported that chitosan particle sizes decreased significantly with increases in the concentrations of bovine serum albumin (entrapped compound). The SEM images further corroborated the TEM results, indicating that the higher concentration of urea (10% *w*/*v*) in chitosan resulted in a decrease in the pore size of the NCUC matrix (Figure 1B), which may be due to the enhanced networking followed by the formation of rodlets and rodlet bundles within the matrix. Similar structural changes in ZnO-CTS-Starch and ZnO-CTS-Whey powder NPs have been observed by [50]. The UV-Vis spectroscopy characterization of the NCUC showed that the incorporation of urea in chitosan shifted the absorption peak from 220 to 200 nm [44]. As urea was incorporated, there was shift in the peak to a lower wavelength, which could be attributed to the formation of chito-urea nanoparticles (Figure 1D). The mechanism for the formation of the chito-urea nanoparticles may include step-wise ionic or hydrogen bond interactions among the linear chitosan chains, sodium TPP and urea. Upon drop-wise addition of sodium TPP in a chitosan-acetic acid solution, the NH_3_^+^ functional group of chitosan exhibited an ionic interaction with the PO_4_^−^ group of tripolyphosphate [51]. The incorporation of urea in a chitosan-acetic acid solution led to its adsorption on the linear chitosan chains through interaction of the amine group(s) in urea with the hydroxyl groups of the chitosan chains [52]. Later, TPP cross-linked the chitosan chains, leading to formation of chito-urea nanoparticles. Previous reports of UV-Vis spectroscopy analysis of chitosan nanoparticles indicated the occurrence of an absorption peak at 217 to 250 nm [53], 226 [54] and 320 [48]. However, a Zn/chitosan NP study provided insights on the blue shift in the absorption peak on increasing the concentration of zinc (Zn) in Zn/chitosan NPs. This possibly indicated the formation of a smaller size of nanoparticles [50], though this may be attributed, partly, to the metallic nature of the ZnO NPs. The major peaks in the FT-IR spectra obtained for the NCUC sample indicated the occurrence of shared peaks for urea and chitosan. The chitosan hydroxyl (O-H) functional group stretching vibrations led to the emergence of peaks at 3450 and 3310 cm^−1^ which have also been observed by Jae-Wook et al. [48]. Likewise, the N-H (amino group) and C-H (alkyl group) bending vibration peaks at 1630 and 1537 cm^−1^ were also identified by them [48].

The release kinetics (Figure 1F) of 1.5% NCUC showed that when the concentration of urea increased, the amount of chitosan might not be sufficient to retain all the urea, due to which some urea molecules might remain on the surface of the chitosan. Likewise, the alpha tocopherol encapsulation study of Luo et al. [55] showed a gradual reduction in the free TOC content released due to the burst effect. It can be inferred that 1.5% NCUC showed the sustainable release of urea within 30 days. The CS and NCUC in vitro biodegradation assay revealed degradation of both the samples, with NCUC exhibiting relatively lowered degradation compared to the CS sample. Likewise, it showed degradation of the chitosan scaffolds [56,57,58], chitosan fibers [59] and chitosan nanoparticles [60].

The results were found to be significant for the soil chemical properties which is consistent with the hypothesis of the slow release of urea from the chitosan matrix (Table 2). The CU-amended soil showed the highest soil organic carbon compared to NCUC. This might be due to chitosan which acted as a carbon source, and it may have contributed to increasing the organic carbon in the soil. Liu et al. [13] observed high OC in an organic fertilizer treated with 400 kg N ha^−1^ compared to a conventional fertilizer treated with 400 kg N ha^−1^. On the other hand, EC was recorded to be the highest in CU followed by NCUC and CS. This observation agrees with the findings of Alva et al. [61] in which the EC of the urea-amended soil was 2- to 10-fold greater than the urea-based CRF treatments. The hydrolysis of urea (without coating) and subsequent transformations were fairly rapid. Higher EC values could potentially decrease the water potential of the soil water and thus inhibit plant growth [62]. However, the EC of the sampled soil under different fertilizer treatments varied from 0.27 to 0.68 dS·m^−1^. These EC values of the tested soil remained lower than the threshold of a saline soil, i.e., EC = 2.4 dS·m^−1^ [63]. Similarly, the pH was recorded to be significantly higher in CU followed by NCUC and CS. The initial increase in the soil pH following the application of urea to the soil was due to hydrolysis of urea into ammonium carbonate, through the action of the urease enzyme [64]. In comparison, the pH values were lower in NCUC, which could be due to the preparation of chitosan-urea nanoformulations in (1% *v*/*v*) acetic acid.

The macronutrient profile of the soil also showed significant variations for the three different types of N sources. Among the different sources of fertilizer applied, CU showed an increased propensity for loss of the applied urea due to the poorly developed root system. The concentration of the ammonical N content was substantially lower in the soil treated with NCUC, which could be due to the controlled urea release property of the chitosan polymer (Figure 3a). Similar studies in cotton involving application of controlled-release urea formulations developed by mixing an epoxy resin coating on sulfur-coated urea and polymer-coated urea had significant positive effects on delaying the NH_3_ volatilization peaks and reducing the overall NH_3_ volatilization in comparison to the urea treatment at the same N application [10]. Likewise, observations on the nitrate contents (Figure 3b) were recorded during the experiment of controlled-release urea, in which the accumulation of nitrate in the soil was significantly lower with polyurethane-coated urea than conventional urea and sulfur-coated urea, which might be due to the controlled release property of the polymer coating [65]. The increase in the concentration of K was supported by Chang et al. [16], who reported an increase in the concentration of the soil K (Figure 3c) content in response to organic fertilizer treatment.

Among the soil microbial populations, the bacterial population at 60 DAT in comparison to 20 and 40 DAT rose significantly (Table 2). This may be due to changes in root secretions that increased the rhizo-deposits and nutrient accessibility to the growing bacteria. These results are in accordance with Daudurand and Knudsen [66], who have reported that bacteria utilize simple amino acids from the rhizosphere of young plants, while they can also utilize complex carbohydrates from the rhizosphere of mature plants, thus indicating that there are modifications in the quality of plant root exudates. Further, the bacterial population can adapt under variable conditions and competition in the niche. The count of ammonia-oxidizing and nitrate-reducing bacteria was lower in NCUC in comparison with CU. The reason behind this can be the slow release of urea from the chitosan polymer, which controlled the N release through the transformation of urea by soil enzymes, besides reducing the emission of N_2_O and diminishing the leaching of urea from the root zone. Similar results have been reported by [67], corroborating that the source of ammonia applied to the soils affects the count and activity of the ammonia oxidizers in the applied soils. A high fungal population was observed in soil treated with NCUC as compared to CU since fungi can flourish in environments having a low N content [68], a condition that is exhibited by application of nanofertilizers owing to their slow nutrient release characteristic [69]. Conversely, CU acted as a rapid source of the carbon nutrient for the growing fungal species, which was observed by a rise in the fungal population from 60 to 90 DAT. The opposite was observed for the bacterial population.

Previous studies have shown that the functional groups involved in N cycling will respond differently to environmental changes, but very little information on their relative responsiveness has been discussed [70]. Ammonia-oxidizing bacteria showed a higher abundance in CU- than NCUC-treated soils. This was due to the increased availability of the ammonia concentration for the CU-treated soil (Figure 4a). The increased concentration of ammonia triggered the process of ammonia oxidation [71]. On the other hand, there was slight variation in the abundance of the AOA population in CU- and NCUC-treated soils (Figure 4b). Similar results were reported by Jung et al. [71], who have observed that AOB were more abundant compared to AOA. Thus, in this study, eubacterial ammonia oxidizers were the most dominant bacterial populations in the nitrification process for the NCUC treatment.

For the soil enzyme activity, CU showed the highest dehydrogenase activity at 60 days at the 100% recommended dose of the fertilizer compared to NCUC and CS. The NCUC showed low urease activity compared to CU, which may be due to the decrease in the availability of urea encapsulated in the chitosan polymer and, thus, lower urea transformation rates which may prevent non-target N losses (Figure 5a). Urease activity increased with an increase in the level of urea application [16,72]. The CU treatment showed the highest enzyme activity at 60 DAT in 100% RDF than the NCUC and CS treatments. The NCUC treatment showed low urease activity compared to the conventional urea treatment. This may be attributed to the presence of the polymeric chitosan matrix which decreased the availability of encapsulated urea and, thus, the transformation of urea, thereby preventing N losses. Similarly, Junejo et al. [17] observed a slowdown in urea hydrolysis in different biodegradable urea coatings in comparison to urea, while the dehydrogenase activity was observed to increase in NCUC and CS treatments. The reason for an increase in the dehydrogenase activity in NCUC and CS could probably be due to chitosan, which may have acted as a degradable carbon source. This extra C source must have led to an enhancement in the number of metabolically active bacteria in these treatments. Previous studies on dehydrogenase activities in maize revealed that CRF (polyolefin-coated urea) significantly increased the soil microbial activity compared to conventional urea [16]. Slow- or controlled-release nitrogen fertilizers substantially affect the total marketable tuber yield in potato besides the N use efficiency [73]. In a three-year field study, Ziadi et al. [74] reported an improved tuber yield of two potato cultivars (Chieftain and Goldrush) by application of commercially available CRU at 150 kg ha^−1^. The researchers argued that this improvement may be attributed to the better availability and uptake of the slowly released N nutrient from the CRF formulation by the root system of the potato plants.

## 5. Conclusions

In this study, encapsulation of urea with the chitosan polymer was carried out, and the final product was characterized through various microscopy and spectroscopy tools. The NCUC product was applied to potato crops under pot conditions. This formulation helped to obtain a potato yield equivalent to the recommended dose (75 kg N ha^−1^) by application of 25% of the required level of N fertilizer. Further, NCUC application significantly reduced the viable counts of soil bacteria (3.36 and 2.02% AOB and NRB, respectively) involved in N transformation and caused a numerical decrease in the urease enzyme activity in the test soil. The soil chemical properties showed significant changes in response to different fertilizer treatments as the pH, EC and OC were negatively and positively influenced by the NCUC treatment. The soil ammonical N and nitrate N contents were low in the NCUC compared to the CU treatment, i.e., 10.69 and 4.55%, respectively. The low concentrations of ammonia and nitrate in the soil supported the slow process of nitrification and controlled release of urea. These results are also corroborated by the reduced population of ammonia-oxidizing and nitrate-reducing bacteria in the soil amended with the NCUC fertilizer. Furthermore, the results show that the relative abundance of AOB and AOA was lower in response to the NCUC treatment. This low abundance of nitrogen-cycling microbial populations in the NCUC-treated soil supports the lower accumulation of nitrate in the soil, and thus its effect on the nitrification and denitrification processes. Thus, the abundance of nitrogen-cycling microbial communities decreased markedly due to the efficient controlled release of the N nutrient by the polymer-encapsulated fertilizer.

## Figures and Tables

**Figure 1 polymers-13-02887-f001:**
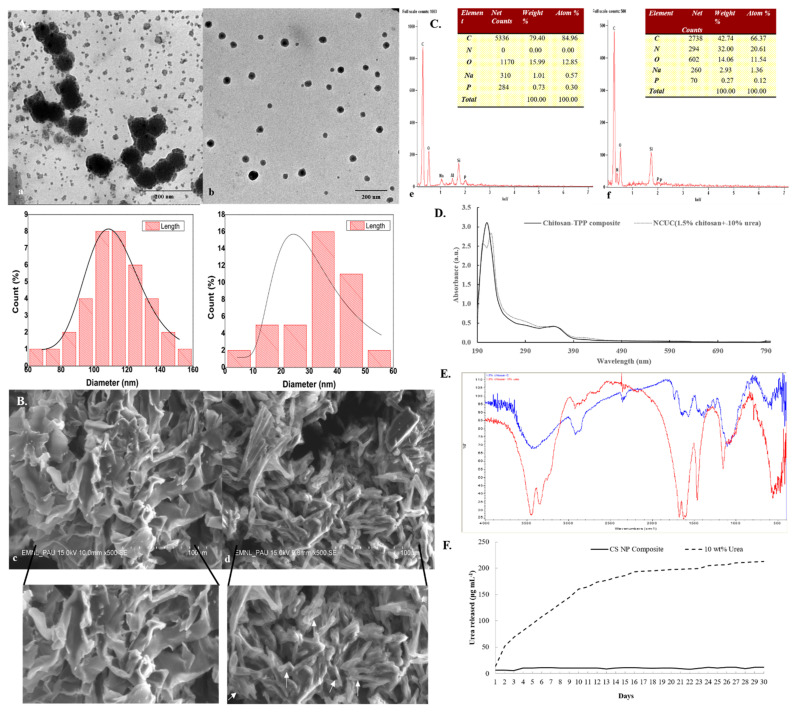
Characterization of chitosan and NCUC. Representative (**A**) TEM micrograph (15.0 K magnification, 80 kV acceleration voltage, high-contrast imaging mode), (**B**) SEM micrograph (500X magnification, 15.0 kV acceleration voltage, secondary electron imaging mode), (**C**) SEM-EDS spectra and elemental composition, (**D**) UV-Vis spectra, (**E**) FT-IR spectra and (**F**) release profile of NCUC. Solid white arrows represent the formation of layered structures in NCUC.

**Figure 2 polymers-13-02887-f002:**
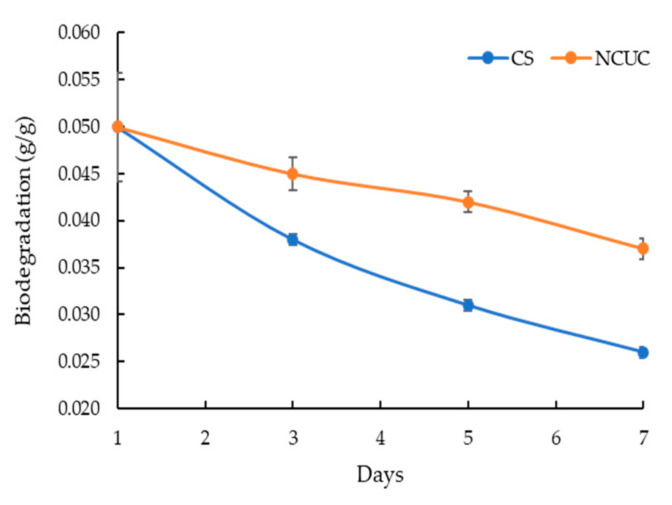
Biodegradation of the chitosan nanoparticles and NCUC samples.

**Figure 3 polymers-13-02887-f003:**
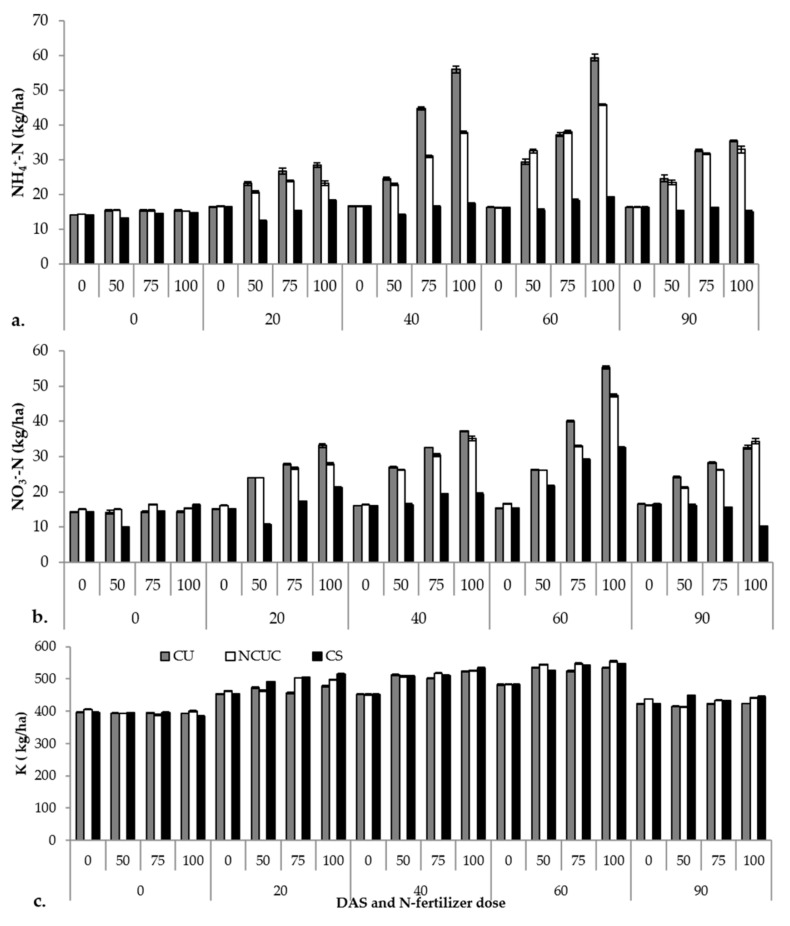
Effect of application of different types of N fertilizers on soil ammonical N (**a**), nitrate N (**b**) and K (**c**) contents of the soil samples under potato cultivation.

**Figure 4 polymers-13-02887-f004:**
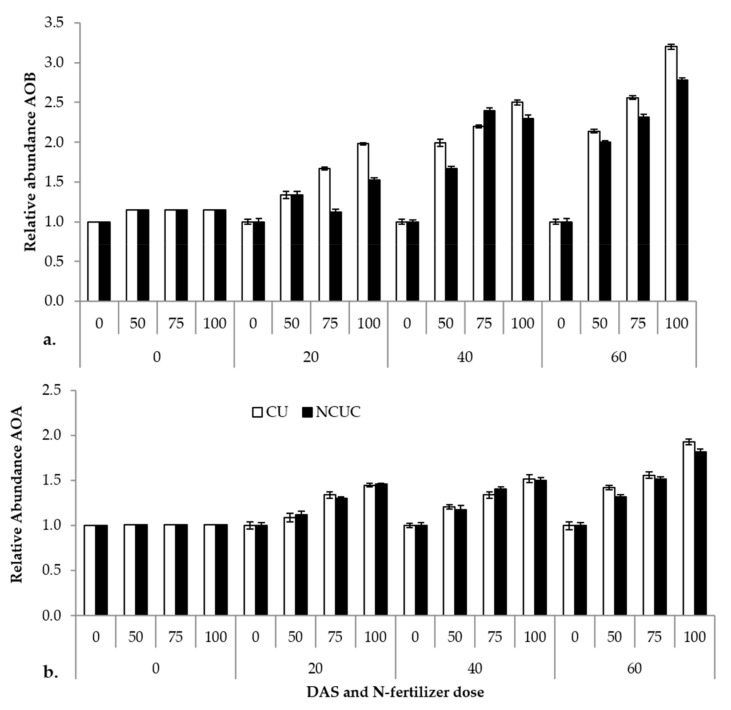
Effect of different types of fertilizer on relative abundance of soil (**a**) ammonia-oxidizing (AOB) and, (**b**) archaeal (AOA) bacteria in potato soil.

**Figure 5 polymers-13-02887-f005:**
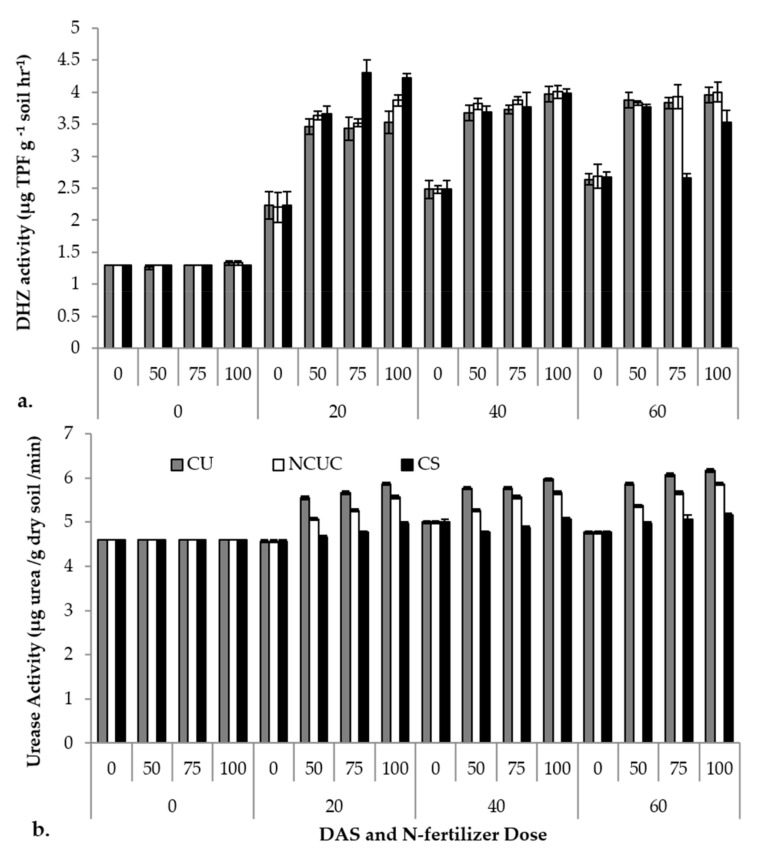
Effect of application of different types of N fertilizers on soil dehydrogenase (**a**) and urease (**b**) activity of the rhizospheric soil of potato cv. Kufri Pukhraj.

**Figure 6 polymers-13-02887-f006:**
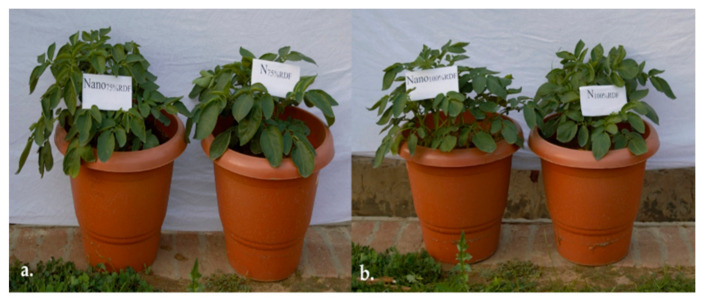
Effect of application of conventional and nano-chitosan-urea composite fertilizer on vegetative growth of potato cv. Kufri Pukhraj: (**a**). N level 75% RDF; (**b**). N level 100% RDF.

**Table 1 polymers-13-02887-t001:** Effect of different types of fertilizer on soil chemical properties and microbial viable cell counts (log cfu g^−1^ soil) during potato cultivation.

Source of Variation	pH	OC (%)	EC (dS/m)	ABC	AOB	NRB	Fungi
DAT							
0	6.50e	0.21e	0.27e	7.45d	4.61e	4.64e	3.83e
20	6.97d	0.27d	0.57d	7.56c	4.83c	4.81c	3.93d
40	7.36b	0.28c	0.59c	7.89b	4.98b	4.92b	4.02c
60	7.62a	0.29a	0.60b	8.02a	5.07a	4.98a	4.14b
90	7.07c	0.28b	0.62a	7.40d	4.78d	4.71d	4.27a
N fertilizer							
CU	7.14a	0.25c	0.54a	7.70b	5.05a	4.95a	4.08b
NCUC	7.12c	0.27b	0.52c	7.81a	4.88b	4.85b	4.16a
CS	7.10b	0.27a	0.53b	7.64c	4.81c	4.76c	4.02c
N level							
0	7.00d	0.23d	0.49d	7.33d	4.64d	4.60d	3.94d
50	7.05c	0.26c	0.51c	7.75c	4.93c	4.86c	4.06c
75	7.16b	0.28b	0.55b	7.85b	5.01b	4.94b	4.14b
100	7.21a	0.29a	0.57a	7.94a	5.08a	5.02a	4.22a

Means within a sub-factor followed by a different letter in a column are significantly different at *p* ≤ 0.05 according to pair-wise comparison of least square means.

**Table 2 polymers-13-02887-t002:** Analysis of variance of days after treatment (DAT), N fertilizer and N level of different types of fertilizers on soil chemical properties during potato cultivation.

SOURCE	DF	pH	OC (%)	EC (dS/m)	ABC	AOB	NRB	Fungi
DAS	4	6.471 ***	0.043 ***	0.788 ***	2.970 ***	0.621 ***	0.501 ***	0.771 ***
Source	2	0.080 ***	0.006 ***	0.006 ***	0.377 ***	0.683 ***	0.397 ***	0.220 ***
DAS*Source	8	0.119 ***	0.005 ***	0.005 ***	0.003 ***	0.003 ***	0.010 ***	0.002ns
N level	3	0.437 ***	0.031 ***	0.054 ***	2.664 ***	1.308 ***	1.218 ***	0.520 ***
DAS*N level	12	0.089 ***	0.002 ***	0.003 ***	0.165 ***	0.027 ***	0.002 ***	0.002ns
Source*N level	6	0.053 ***	0.007 ***	0.001 ***	0.044 ***	0.076 ***	0.051 ***	0.026 ***
DAS*Source*N level	24	0.022 ***	0.001 ***	0.0001 ***	0.002 ***	0.002 ***	0.002 ***	0.001ns

*** = *p* ≤ 0.001, ns = non-significant. DF: degree of freedom, OC: organic carbon, EC: electrical conductivity, ABC: aerobic bacterial count, AOB: ammonia-oxidizing bacteria, NRB: nitrate-reducing bacteria.

**Table 3 polymers-13-02887-t003:** Effect of different sources and concentrations of fertilizers on shoot fresh and dry weight (g) parameters.

N Fertilizer Level	Fresh wt.			Dry wt.			Mean
	CU	NCUC	CS	CU	NCUC	CS	
0	18.27 ± 0.10d	18.27 ± 0.10d	18.27 ± 0.10d	15.27 ± 0.14d	15.00 ± 0.25d	15.00 ± 0.25d	16.68
50	23.39 ± 0.36c	28.63 ± 0.12c	23.50 ± 0.21c	19.67 ± 0.19c	24.63 ± 0.17c	18.53 ± 0.19c	23.06
75	26.48 ± 0.03b	31.64 ± 0.21b	24.54 ± 0.23b	23.33 ± 0.06b	28.67 ± 0.12b	19.67 ± 0.15b	25.72
100	27.44 ± 0.41a	35.70 ± 0.08a	26.33 ± 0.38a	24.93 ± 0.19a	32.40 ± 0.12a	23.27 ± 0.20a	28.34
Mean	23.89	28.56	23.16	20.8	25.17	19.12	

Means within a sub-factor followed by a different letter in a column are significantly different at *p* ≤ 0.05 according to a pair-wise comparison of least square means.

## Data Availability

The data are included in the article.

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
