# Peer review of "Chitosan-Urea Nanocomposite for Improved Fertilizer Applications: The Effect on the Soil Enzymatic Activities and Microflora Dynamics in N Cycle of Potatoes (Solanum tuberosum L.)"

_polymers, 2021, doi:10.3390/polym13172887_

Round 1

Reviewer 1 Report

The manuscript presents data on preparation and properties of a nano-chitosan-urea composite (NCUC) fertilizer. Efficiency of NCUC was demonstrated in terms of its effects on various soil properties (pH, OC, EC, content of NH4+ and NO3-, abundance of AOA and AOB genes, dehydrogenase and urease activity). Data on potato yield after NCUC application at different rates in comparison with traditional urea is also presented. Substances under study were characterized using UV-Vis and FT-IR spectroscopy, scanning and transmission electron microscopy. The manuscript looks like an interesting applied research related to the production and characterization of slow-release fertilizers.

Comments and remarks

  1. Line 65. PCU is not a novel fertilizer. Please correct.
  2. Line 88. Please add a short explanation what benefits of chitosan were expected as compared to the other coating agents. Why chitosan has been selected for the study?
  3. Line 175. Please give a name of the soil used according WRB rather that USDA system of classification.
  4. Line 201. Please add a description for NH4+ and NO3- determination.
  5. Lines 228-234. Please add references where the listed primers were recommended for AOB and AOA estimation.
  6. Line 242. Please specify what a “marketable tuber” means.
  7. Lines 285-291. Here the authors describe data, which are not presented in the manuscript. Please add a Table with the data on pH, OC, EC. Please change SOC for OC.
  8. Table 1. Please decipher DF, ABC, and NRB. How ABC and NRB were measured? It looks illogical to present ANOVA results (Table 1) before data subjected to the analysis (Table 2). May be the authors want to change the order of Table 1 and 2.
  9. Table 2. The meaning of the presented data is unclear. For example, for pH and CU the value is 7.14. Does this value correspond to the value of pH for the variant with CU introduction at DAS 0? At what dosage? Or this value is the average for all values of pH where CU was applied? Please explain. The way how the data were presented in Supplementary Table 1 looks more understandable. May be the authors want to present all the data subjected to ANOVA in this way and present them in Supporting information. Please add information about pair-wise comparison of least square means to the sub-section 2.14.
  10. Line 343. Please check formatting.
  11. Figure 6. Please use CU and NCUC instead of Conv and Nano.
  12. Line 413. Please change SOC for soil OC, as it was OC that was introduced as an abbreviation (line 196).
  13. Line 420. According Table 2 EC was changed due to fertilization. So, please specify what is the difference between “sever” and “not severe” alterations in the soil EC.
  14. Lines 462-463. Here is a repetition of 327-329. Please delete.
  15. Line 492. Please add a short discussion of CU and NCUC on potato yield.
  16. Line 494-495. Is difficult to find data in the manuscript that would confirm this statement. Figure 6 demonstrates that there was no statistically significant difference between effcets for CU and NCUC. Supplementary Table 2 contains results of statistical data treatment demonstrating difference between CU and NCUC. However, data in this table does not indicate what variant demonstrated higher yield. So, a table containing data about the yield after CU and NCUC application at different dosage is of need.
  17. Line 32. There is no data on biodegradability of NCUC.

Author Response

Dear Reviewer,

We sincerely thank you for the critical reading and for your constructive suggestions which have immensely helped to improve the quality of the manuscript.

We hope that you will be satisfied by the point-wise replies that have been attached alongwith this email.

with best regards

authors

Reviewer 2 Report

General comments

The present manuscript is focused on the preparation and characterisation of nano-chitosan-urea composite (NCUC) to be used as fertilizer for the treatment of soil, investigating its effect on soil enzymatic activities and microflora involved in N-cycling of potato (Solanum tuberosum L.).

The topic of paper is interesting and worthy of investigation. The manuscript is well organised and conceived.

However, the suggested major revisions have to be strictly applied before considering this paper for publication.

Specific suggestions are reported below point by point.

In addition to a deep and accurate revision of the English language, some suggestions are reported below point by point.

Keywords

  • The chosen keywords (i.e. Ammonia oxidase gene; microflora N-cycle; nanocomposite; nutrient use efficiency; quantitative PCR) do not completely cover the manuscript content. Please add some keywords about the used materials, properties and applications.
  • It is not so usual to use acronym, such as PCR, among the keywords. Please replace it with its extended name.
  • Moreover, the keywords have to be reported in a logical order (i.e. material, processing, characterisations, properties, applications).

  1. Introduction

- The Introduction section is well conceived and organised. It is clear and well highlights bot the aim and the originality of the present paper.

- The Authors have only to justify the choice of chitosan as urea coating and describe the properties and applications of chitosan, citing proper references, such as “A comprehensive review on the nanocomposites loaded with chitosan nanoparticles for food packaging, Critical Reviews in Food Science and Nutrition 2020, https://doi.org/10.1080/10408398.2020.1843133”,d “Clay/chitosan biocomposite systems as novel green carriers for covalent immobilization of food enzymes, Journal of Materials Research and Technology 8[4] (2019): 3644-3652.”, “Screen-printed electrode modified with carbon black nanoparticles and chitosan: a novel platform for acetylcholinesterase biosensor development, Analytical & Bioanalytical Chemistry 408(2016): 7299-7309” and “In situ temperature sensing with fluorescent chitosan-coated PNIPAAm/alginate beads, Journal of Materials Science 52[20] (2017): 12506–12512”, for food, sensing and optical properties.

  1. Materials and Methods

2.1 Materials

- More details about all the used reagents and solvents have to be added, such as the molecular weight (for chitosan the related medium MW has to be specified) and the purity.

2.2. Synthesis and characterization of nano-polymer urea composite

- The acetic acid concentration has to be specified. Usually chitosan is solubilised in and acetic acid water solution.

- How was TPP added to urea-chitosan solution? Drop wise? Please specify the drop rate.

  • The following statement “Instantaneously, the synthesis of chitosan-urea nanoparticles starts due to the TPP-initiated ionic gelation mechanism.” has to be supported with suitable references, including “Biodegradable zein film composites reinforced with chitosan nanoparticles and cinnamon essential oil: physical, mechanical, structural and antimicrobial attributes, Colloids and Surfaces B: Biointerfaces 177(2019): 25-32”.

2.3. UV-Vis spectroscopy

-The resolution has to be reported.

2.5. Scanning electron microscopy (SEM) and SEM-Energy dispersive spectroscopy (SEM-EDS)

- The applied current and time for the gold sputtering have to be specified.

2.6. Fourier Transform-Infra Red Spectroscopy (FT-IR Spectroscopy)

- The KBr/powder ratio has to be specified.

2.9. Soil chemical characterization

- Concerning the macronutrient status (N, 200 P and K) of the soil, the followed procedure has to be briefly described, even if reported elsewhere.

2.11. Soil enzyme activity

- The protocols followed for the dehydrogenase and urease activities have to be briefly described, even if reported elsewhere.

  1. Results

3.1. Characterization of Nano-polymer urea composite

- The revealed peak at 220 nm has to be clearly ascribed and explained.

- The following statement “This indicates decrease in the size of the synthesized nano-chitosan particles and encapsulation of urea in the chitosan matrix” has to be supported with suitable references.

- How were the dimensions evaluated by TEM images? Please add details in the related experimental section, specifying the used software and the number of measured particles to provide an average value and a standard deviation.

- The shown TEM images have to be better and more deeply described.

- The reported SEM micrographs are very similar; thus one can be removed, since the other one does not provide further information.

- The following phrase “The FT-IR spectra of the chitosan showed presence of characteristic absorption peak at 16 and NCUC (Figure 1E) demonstrated occurrence of specific bands at 1550-1650 cm-1, 1600-1700 cm-1 and 2900 cm-1, respectively” has to be rewritten.

- The reported FTIR spectra have to better and more deeply described, properly assigning all the detected peaks with appropriate references. For chitosan, the Authors could use and cite the reference “Chitosan/clay nanocomposite films as support for enzyme immobilization: an innovative green approach for reducing the haze potential in white wine, Food Hydrocolloids 74(2018): 124-131”.

  1. Discussion

- All this period “The FT-IR analysis 398 (Figure 1E) revealed the occurrence of peaks at 1634 cm-1 in chitosan associated with the stretching vibrations of NH-CO, C=O along with bending bonds of N-H and NH2 [36], whereas incorporation of urea in chitosan showed sharp and narrow peaks at 1620 cm-1 and 1461 cm-1, which are due to amide II bending and C–N stretching vibrations [31]” cannot be considered a discussion, but it is a description. Thus, it could be moved to the Results section.

- In the following sentence “The increase in the concentration of K was supported by Chang et al. (2007) who have reported an increase in the concentration of soil K (Figure 2c) content in response to organic fertilizer treatment.” please add the related numbered reference ([16]).

  1. Conclusions

- A contextualization has to be added.

- The main significant numerical data have to be reported.

Author Response

Dear Reviewer,

We sincerely thank you for your critical comments and constructive suggestions.

The point-wise relies to the comments are appended below:-

Reviewer 2:

Comment 1: The present manuscript is focused on the preparation and characterisation of nano-chitosan-urea composite (NCUC) to be used as fertilizer for the treatment of soil, investigating its effect on soil enzymatic activities and microflora involved in N-cycling of potato (Solanum tuberosum L.). The topic of paper is interesting and worthy of investigation. The manuscript is well organised and conceived. However, the suggested major revisions have to be strictly applied before considering this paper for publication. Specific suggestions are reported below point by point. In addition to a deep and accurate revision of the English language, some suggestions are reported below point by point.

Reply: Sincere thanks to the reviewer for appreciating the research concept of the manuscript. We are also thankful for the valuable insight and suggestions. The replies to the specific comments have been given in point-wise manner.

 Comment 2: Keywords

Comment 2a: The chosen keywords (i.e. Ammonia oxidase gene; microflora N-cycle; nanocomposite; nutrient use efficiency; quantitative PCR) do not completely cover the manuscript content. Please add some keywords about the used materials, properties and applications.

Reply: New keywords have been added in the revised manuscript.  

Comment 2b: It is not so usual to use acronym, such as PCR, among the keywords. Please replace it with its extended name.

Reply: The extended forms for the word PCR has been incorporated in the revised version.

Comment 2c: Moreover, the keywords have to be reported in a logical order (i.e. material, processing, characterisations, properties, applications).

Reply: New keywords have been added as per the suggestions and the keywords are kept in alphabetical order.

Comment 3: Introduction

Comment 3a: The Introduction section is well conceived and organized. It is clear and well highlights bot the aim and the originality of the present paper.

The Authors have only to justify the choice of chitosan as urea coating and describe the properties and applications of chitosan, citing proper references, such as “A comprehensive review on the nanocomposites loaded with chitosan nanoparticles for food packaging, Critical Reviews in Food Science and Nutrition 2020, https://doi.org/10.1080/10408398.2020.1843133”,d “Clay/chitosan biocomposite systems as novel green carriers for covalent immobilization of food enzymes, Journal of Materials Research and Technology 8[4] (2019): 3644-3652.”, “Screen-printed electrode modified with carbon black nanoparticles and chitosan: a novel platform for acetylcholinesterase biosensor development, Analytical & Bioanalytical Chemistry 408(2016): 7299-7309” and “In situ temperature sensing with fluorescent chitosan-coated PNIPAAm/alginate beads, Journal of Materials Science 52[20] (2017): 12506–12512”, for food, sensing and optical properties.

Reply: The indicated manuscripts have been cited in the revised manuscript in the introduction section.

Comment 4: Materials and Methods

Comment 4a: Materials: More details about all the used reagents and solvents have to be added, such as the molecular weight (for chitosan the related medium MW has to be specified) and the purity.

Reply: The molecular weight and purity of the chitosan used in the study has been incorporated in the revised version of the manuscript.

Comment 4b(i): Synthesis and characterization of nano-polymer urea composite: The acetic acid concentration has to be specified. Usually, chitosan is solubilised in and acetic acid water solution.

Reply: The acetic acid concentration has been added in M&M section.

Comment 4b(ii): How was TPP added to urea-chitosan solution? Drop wise? Please specify the drop rate.

Reply: The TPP was added to the chitosan-urea solution in drop-wise manner with drop rate of 16 drops per minutes (⁓2 mL per minute).

Comment 4b(iii): The following statement “Instantaneously, the synthesis of chitosan-urea nanoparticles starts due to the TPP-initiated ionic gelation mechanism.” has to be supported with suitable references, including “Biodegradable zein film composites reinforced with chitosan nanoparticles and cinnamon essential oil: physical, mechanical, structural and antimicrobial attributes, Colloids and Surfaces B: Biointerfaces 177(2019): 25-32”.

Reply: The suggested reference and three other papers having description of spontaneous or instant synthesis of chitosan NPs have been cited at the indicated place in the revised manuscript.

Comment 4(c):2.3. UV-Vis spectroscopy: The resolution has to be reported.

Reply: The specification document of the UV Vis spectrometer has the following parameters that affect the spectral resolution i.e. Bandwidth: 1.8 nm, Readability: 0.1 nm, Accuracy: ± 0.5 nm and Repeatability: ± 0.2 nm.

Comment 4(d): 2.5. Scanning electron microscopy (SEM) and SEM-Energy dispersive spectroscopy (SEM-EDS): The applied current and time for the gold sputtering have to be specified.

Reply: The applied current and time details (18–20 mA for 30 seconds) for the gold sputtering have been provided in the revised version.

Comment 4(e): 2.6. Fourier Transform-Infra Red Spectroscopy (FT-IR Spectroscopy): The KBr/powder ratio has to be specified.

Reply: The KBr/powder ratio has been 100:1 and has been mentioned in the revised version of the manuscript. 

Comment 4(f): 2.9. Soil chemical characterization:  Concerning the macronutrient status (N, 200 P and K) of the soil, the followed procedure has to be briefly described, even if reported elsewhere.

Reply: The brief description of the procedures has been incorporated in the revised version of the manuscript. 

Comment 4(g): 2.11. Soil enzyme activity: The protocols followed for the dehydrogenase and urease activities have to be briefly described, even if reported elsewhere.

Reply: The brief description of the protocols of the urease and dehydrogenase enzyme activities has been added in the revised manuscript.

Comment 5: Results: 3.1. Characterization of Nano-polymer urea composite

Comment 5(a): The revealed peak at 220 nm has to be clearly ascribed and explained.

Reply: The revealed peak at 220 nm corresponds to characteristic absorption peak for chitosan (226 nm) as reported by AbdElhady et al. (2012).  

AbdElhady, M.M. Preparation and Characterization of Chitosan/Zinc Oxide Nanoparticles for Imparting Antimicrobial and UV Protection to Cotton Fabric. Int. J. Carbohydr. Chem. 2012, 2012, 1–6, doi:10.1155/2012/840591.  

Comment 5(b): The following statement “This indicates decrease in the size of the synthesized nano-chitosan particles and encapsulation of urea in the chitosan matrix” has to be supported with suitable references.

Reply: Pertinent literature has been incorporated to support the modified statement in the revised manuscript.    

Comment 5(c): How were the dimensions evaluated by TEM images? Please add details in the related experimental section, specifying the used software and the number of measured particles to provide an average value and a standard deviation.

Reply: The dimensions of the chitosan and chitosan-urea nanoparticles were measured through Image J software (version 1.46r, NIH, USA) by manually obtaining the diameters for >50 particles from five TEM micrographs.

Comment 5(d): The shown TEM images have to be better and more deeply described.

Reply: The description of the TEM image has been improved in the revised version of the manuscript.

Comment 5(e): The reported SEM micrographs are very similar; thus one can be removed, since the other one does not provide further information.

Reply: The difference in the two SEM images have been depicted in insets in the revised Figure 1. There is formation of layered structures on incorporation of the urea (10% w/v) in the chitosan. The layered structures have been indicated with solid white arrows in the image as well as in the inset of NCUC product.

Similar fibrous structure formation has been reported by Nie et al. (2016) through SEM and Confocal studies of the metal-chitosan hydrogel.

Nie, J., Wang, Z. & Hu, Q. Chitosan Hydrogel Structure Modulated by Metal Ions. Sci Rep 6, 36005 (2016). https://doi.org/10.1038/srep36005

Comment 5(f): The following phrase “The FT-IR spectra of the chitosan showed presence of characteristic absorption peak at 16 and NCUC (Figure 1E) demonstrated occurrence of specific bands at 1550-1650 cm-1, 1600-1700 cm-1 and 2900 cm-1, respectively” has to be rewritten.

Reply: The suggested sentence has been revised.

Comment 5(g): The reported FTIR spectra have to better and more deeply described, properly assigning all the detected peaks with appropriate references. For chitosan, the Authors could use and cite the reference “Chitosan/clay nanocomposite films as support for enzyme immobilization: an innovative green approach for reducing the haze potential in white wine, Food Hydrocolloids 74(2018): 124-131”.

Reply: The description has been shifted from the discussion section to the results section. The indicated reference has been cited in the revised manuscript.

Comment 6: Discussion

Comment 6(a): All this period “The FT-IR analysis 398 (Figure 1E) revealed the occurrence of peaks at 1634 cm-1 in chitosan associated with the stretching vibrations of NH-CO, C=O along with bending bonds of N-H and NH2 [36], whereas incorporation of urea in chitosan showed sharp and narrow peaks at 1620 cm-1 and 1461 cm-1, which are due to amide II bending and C–N stretching vibrations [31]” cannot be considered a discussion, but it is a description. Thus, it could be moved to the Results section.

Reply: The indicated description has been moved to the results section.

Comment 6(b): In the following sentence “The increase in the concentration of K was supported by Chang et al. (2007) who have reported an increase in the concentration of soil K (Figure 2c) content in response to organic fertilizer treatment.” please add the related numbered reference ([16]).

Reply: The related number of the reference has been incorporated in the revised manuscript.

Comment 7: Conclusions

Comment 7(a): A contextualization has to be added.

Reply: The conclusion section has been revised as per the suggestion.

Comment 7(b): The main significant numerical data have to be reported.

Reply: The significant numerical data has been incorporated in the conclusion section of the revised manuscript.

Round 2

Reviewer 1 Report

The authors provide adequate responses to points raised by the reviewer. However, not all required changes were made within the actual manuscript. It is likely other readers will have similar questions and therefore the manuscript would improve by adding the answers. More specifically, please add in the manuscript or in the Supplementary the answers concerning:

– Comment 7 (definition of the “marketable tuber”);

– Comment 9 (procedure of ABC and NRB estimation);

– Comment 14 (explanation of the term “no severe alteration”);

– Comment 18 (data on biodegradation of chitosan).

Author Response

Dear Reviewer,

Thanks for your suggestions and comments. The point-wise replies to the comments have been appended below:-

Comment 7: (definition of the “marketable tuber”);

Reply: The definition of the marketable tuber has been added in the materials and methods section 2.8.

Comment 9: (procedure of ABC and NRB estimation);

Reply: The procedure of ABC and NRB estimation has been incorporated in the materials and methods section 2.11 in the revised manuscript.

Comment 14: (explanation of the term “no severe alteration”);

Reply: The explanation for no severe alteration has been incorporated by modifying the sentence in the discussion section of the manuscript.

Comment 18: (data on biodegradation of chitosan).

Reply: The degradation of the chitosan urea composite has been incorporated in the new materials and methods, results, and discussion sections. 

Sincere thanks and with best regards

anu

Reviewer 2 Report

General comments

The Authors have almost followed all the Referees’remarks. Some minor revisions have to be applied before considering for publication in Polymers, as reported below.

Keywords

  • As already requested, the keywords have to be reported in a logical order (i.e. material, processing, characterisations, properties, applications).

  1. Materials and Methods

2.1 Materials

- For some reagents, the purity has not been specified yet.

Author Response

Dear Reviewer,

Sincere thanks for your comments and suggestions.

The same have been incorporated in the revised manuscript. The point-wise replies are appended below:-

Comment 1: As already requested, the keywords have to be reported in a logical order (i.e. material, processing, characterisations, properties, applications).

Reply: The keywords have been reordered as per the logical order as suggested.

Comment 2: Materials and methods section: For some reagents, the purity has not been specified yet.

Reply: The purity of the remaining reagents has been incorporated in the revised manuscript.

with best regards

anu